# Emergent dipole field theory in atomic ladders

Hernan B. Xavier[1,2]⋆, Poetri Sonya Tarabunga[1,2,3], Marcello Dalmonte[1], and Rodrigo G. Pereira[4]

**1** The Abdus Salam International Centre for Theoretical Physics (ICTP), Trieste, Italy
**2** International School for Advanced Studies (SISSA), Trieste, Italy
**3** INFN, Sezione di Trieste, Via Valerio 2, 34127 Trieste, Italy
**4** International Institute of Physics and Departamento de Física Teórica e Experimental,
Universidade Federal do Rio Grande do Norte, Natal-RN, Brazil

⋆ hxavier@ictp.it

## Abstract

We study the dynamics of hard-core bosons on ladders, in the presence of strong kinetic constrains akin to those of the Bariev model. We use a combination of analytical methods and numerical simulations to establish the phase diagram of the model. The model displays a paired Tomonaga-Luttinger liquid phase featuring an emergent dipole symmetry, which encodes the local pairing constraint into a global, nonlocal quantity. We scrutinize the effect of such emergent low-energy symmetry during quench dynamics including single-particle defects. We observe that, despite being approximate, the dipole symmetry still leads to very slow relaxation dynamics, which we model via an effective field theory. The model we discuss is amenable to realization in both cold atoms in optical lattices and Rydberg atom arrays with dynamics taking place solely in the Rydberg manifold. To observe the unusual dynamics of excitations in such experimental platforms, we propose a two-step protocol, which starts with the quasi-adiabatic preparation of low-energy states, followed by the local creation of defects and their study under quench dynamics.

# 1   Introduction

Sparked by a series of remarkable atomic physics experiments [1–4], constrained dynamics in quantum many-body systems has attracted a great deal of attention in recent years [5–16]. From a fundamental viewpoint, these systems offer a rich playground for studying complex non-equilibrium properties, where the interplay of correlations and dynamical frustration can result in a variety of elusive phenomena, such as Hilbert space fragmentation [17–21], slow relaxation dynamics [22–24], as well as intriguing links to lattice gauge theories [25–28] and fracton models [29–31]. Such phenomena are inherently related to strong correlations and lack a counterpart in the context of noninteracting particles.

The imposition of higher, multipole conservation laws in many-body theory [6,27] represents a new opportunity to generate nontrivial dynamics. This has been recently demonstrated in experiments with ultracold atom gases [2–4], where a tilted optical lattice is used to enforce a dipole (center-of-mass) preserving dynamics. In view of that, a number of studies have not only established the ground state phase diagram of dipole-conserving lattice models of fermions or bosons [12], but have also considered the nonequilibrium dynamics of such models [14,15]. The core idea about this research line is to enforce as well as possible the dipole symmetry as a Hamiltonian property (regardless of energy scale).

In this work we pursue a different approach, where the symmetry constraint emerges as a low-energy property of the ground state. We draw inspiration from a Bariev-like model [32–36], whose phase diagram features a Tomonaga-Luttinger liquid (TLL) state formed by bound pairs. We use field theory arguments and exact diagonalization (ED) to link this ground state to an emergent dipole-type symmetry, which constrains the local dynamics of single-particle excitations that need to find partners in order to move. Having in mind atomic physics realizations (Fig. 1), we propose a quasi-adiabatic protocol [37] to prepare the dipole TLL state from biased optical ladders, as well as Rydberg atom arrays. We benchmark the state preparation by using a time-dependent variational principle (TDVP) algorithm to dynamically evolve an initial product state. Finally we consider the quench dynamics of isolated defects placed on top of the dipole TLL state. We use a combination of density matrix renormalization group (DMRG) and time-evolving block-decimation (TEBD) algorithms to prepare and evolve the single-defect states, contrasting our numerical results to field theory predictions. Figure 1 gives a general perspective onto a protocol that prepares and observes the time evolution of defects.

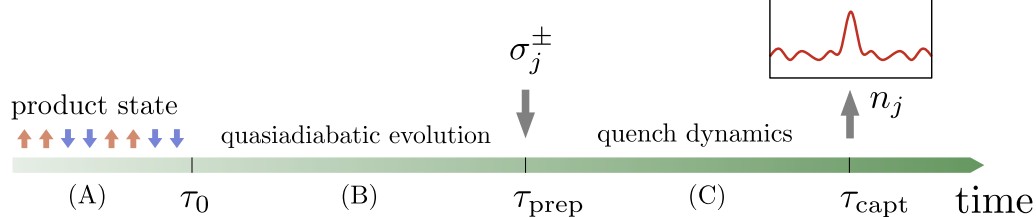

Figure 1: Schematic of the protocol to observe fragmentation dynamics in the presence of an emergent dipole symmetry. (A) Initially, a product state is prepared in the first time interval. (B) This is followed by a quasi-adiabatic state preparation over a time frame $T = \tau_{\mathrm{prep}} - \tau_0$. (C) Once the target state is prepared, a single-particle defect is created by the action of the local operator $\sigma_j^{\pm}$, and the resulting state is left to undergo unitary evolution during the time interval. At the final time the density $n_j$ is measured.

The remainder of the paper is organized as follows. We set our notation and introduce the model Hamiltonian as a hardcore boson ladder in Sec. 2. We inspect analytically specific parameter regimes, and then focus on the regime where the model exhibits strictly confined excitations. We utilize a duality map to reveal a connection to a PXP-type model featuring Hilbert space fragmentation, providing an alternative viewpoint on dipole symmetry at one of the exactly solvable points the model features. We discuss experimental implementations of the microscopic dynamics in Sec. 3: we derive the effective Hamiltonian from physically sensible Rydberg- and cold-atom models, discussing pertinent perturbations to each platform. A low-energy field theory description for the lattice model is presented in Sec. 4, which we use to assess the stability of the dipole TLL state and devise the mobile impurity model. This analysis is complemented by numerical simulations in Sec. 5. We give particular focus to benchmark the state preparation protocol and test the nontrivial dynamics of isolated defects. We offer a summary and point out open perspectives in Sec. 6.

## 2   Effective hard-core boson model

We consider hard-core bosons on a two-leg zigzag ladder (shown in Fig. 2), where the number of particles is preserved separately in each sublattice. The system dynamics is described by a Bariev-like [32, 36] Hamiltonian:

$$H = -J \sum_i (\sigma_i^+ \sigma_{i+2}^- + \mathrm{H.c.}) - W \sum_i (\sigma_i^+ \sigma_{i+1}^z \sigma_{i+2}^- + \mathrm{H.c.}) + V \sum_i \sigma_i^z \sigma_{i+1}^z, \qquad (1)$$

where $i = 1, 2, \ldots, L$ are sites in the zigzag geometry, and $\sigma_i^+$ is the hard-core boson creation operator. The hard-core boson occupation number $n_i = \sigma_i^+ \sigma_i^-$ is related to the Pauli matrix $\sigma_i^z$ by $n_i = (1 + \sigma_i^z)/2$. The lattice model depends on three coupling parameters. The intra-leg hopping amplitude $J$, the correlated hopping term $W$, and the Ising-like interaction $V$. Note that single-particle tunneling between different legs is not allowed as it violates the number conservations.

The two global U(1) symmetries of the model can be expressed as the conservation of the total number of excitations and the so called inter-leg *magnetization*:

$$N = \sum_i n_i, \qquad M = \frac{1}{2} \sum_i (-1)^i n_i. \qquad (2)$$

We call $N$ and $M$ the *charge* and *spin* quantum numbers, given the similarity to electronic systems. We use these two conserved charges to fix a subspace of interest. Hereafter, we concen-

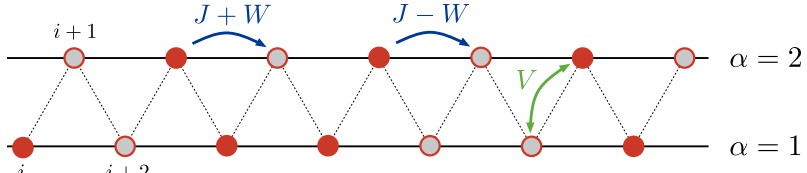

Figure 2: Illustration of the hard-core boson model. The two-leg ladder has the geometry of a zigzag chain where odd and even sub-lattices enjoy their own number conservation. Correlated hopping $W$ modulates tunneling amplitudes according to the presence or absence of bosons in the other leg. Additional nearest-neighbor interactions $V$ favor attraction or repulsion among neighboring bosons.

trate on the scenario where both legs are half-filled, meaning we shall consider chains whose thermodynamic limit is defined by $N/L \to 1/2$, with vanishing magnetization $M/L \to 0$. We use $J = 1$ to set the energy scale, and study the model as a function of $W$ and $V$. We only consider positive values of $W$, since we can flip its sign via a global particle-hole transformation, $\sigma_i^z \to -\sigma_i^z$.

## 2.1 Phase diagram overview

Bariev-like models and their corresponding phase diagrams have been discussed in the literature [32–36]. Below, we exploit some of these earlier results to clarify the phase diagram of the hard-core boson ladder, referring the reader to Refs. [35, 36] for more details. We will complement those with a field theory approach we describe below.

The phase diagram is schematically depicted in Fig. 3. At $V = W = 0$, we find the 2TLL phase, a critical state with power-law decaying correlation functions described by two independent TLL theories (as for decoupled chains). The TLLs govern the low-energy spectrum of collective excitations of charge and spin that are generated from the hybridization of the original excitations in the legs. This phase arises from the competition among the correlated hopping $W$ and the antiferromagnetic $V$ interaction that prevents excitations from pairing up.

If we keep $V$ small and move towards dominant $W$ coupling, spin excitations are gapped out and we enter the dipole TLL phase. The dipole TLL phase can be viewed as a quantum liquid of *molecular* dimers [32], where each dimer is formed by binding two single-particle excitations that live in distinct legs together. The associated pairing strength depends mainly on the correlated hopping $W$ [33], which drives a BKT-type transition at $V = 0$ into the dipole TLL phase. The underlying liquid ground state can be identified by a den Nijs-Rommelse-type string order parameter, as shown by Chhajlany *et al.* [36].

Alternatively, we identify here an emergent dipole-like symmetry that constrains the dynamics of single-particle excitations in the dipole TLL state. At $W = J$ the binding strength reaches its maximum [33]. At this special point, excitations are strictly confined (see Fig. 4) and the dipole symmetry becomes exact as the lattice model Eq. (1) commutes with

$$D = \sum_i i\tilde{n}_i, \qquad \tilde{n}_i = n_i \exp\Big(i\pi \sum_{j<i} n_j\Big), \qquad (3)$$

which encodes the local constraint in a nonlocal global operator. We point out that, although clearly grounded in the fixed points provided by the Bariev chain obtained for $V = 0$ [32], the dipole TLL phase is quite robust to a great sort of perturbations, cf. Sec. 3, surviving up to finite values of $V$ [36].

Finally, for dominant $V$ coupling we find the phase-separated (PS) states $PS_c$ and $PS_s$. The phases $PS_c$ and $PS_s$ correspond to phase separation of charge and ladder spin [35,38]. We note

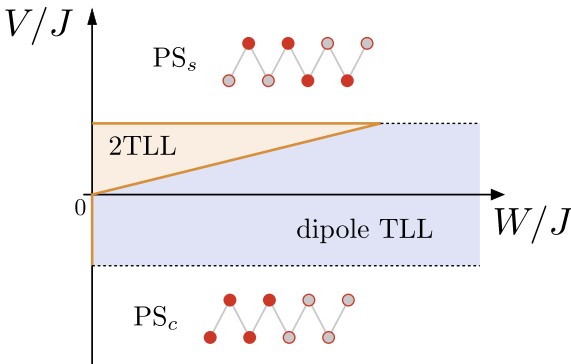

Figure 3: Simplified sketch of the phase diagram as obtained from the weak-coupling theory. Near the origin we find the 2TLL and dipole TLL phases. For large $V$ these give room to phase separation $PS_c$ and $PS_s$.

these are particularly sensitive to boundary conditions (as well as form of interactions, e.g., replacing Ising-like to density-density interactions, which involve fixing the chemical potential, leaving behind extra boundary fields for open geometries). For simplicity we only comment on open chains with a double even number of sites, so each sublattice is at exact half filling.

Ferromagnetic interactions favor clustering, giving rise to the $PS_c$ phase at large enough $V < 0$. The two degenerate classical ground states of the Ising-like interaction take the domain-wall form

$$|\phi_1\rangle = |\bullet\bullet\bullet\bullet\bullet\bullet\circ\circ\circ\circ\circ\circ\rangle, \qquad |\phi_2\rangle = |\circ\circ\circ\circ\circ\circ\bullet\bullet\bullet\bullet\bullet\bullet\rangle, \qquad (4)$$

breaking $\mathbb{Z}_2$ mirror reflection across the center link of the chain. The gapped spectrum is composed from the creation of additional domain-walls, obtained by either the full displacement of the cluster or its separation in smaller pieces. The addition of the couplings $J$ and $W$ do not lift the twofold degeneracy of the $PS_c$ ground state, but give kinetic energy to domain-wall excitations renormalizing energy gaps.

On the other hand, strong antiferromagnetic interactions lead to the so-called $PS_s$ phase, which can be thought of as a Néel state with a domain-wall excitation stuck in the center of the chain [35]. As in the ferromagnetic case, the antiferromagnetic Ising-like interaction has two classical ground states:

$$|\psi_1\rangle = |\circ\bullet\circ\bullet\circ\bullet\bullet\bullet\circ\bullet\circ\bullet\circ\rangle, \qquad |\psi_2\rangle = |\bullet\circ\bullet\circ\bullet\circ\circ\bullet\circ\bullet\circ\bullet\rangle. \qquad (5)$$

However, while the addition of $J$ remains innocuous, a nonzero $W$ lifts the ground state degeneracy. The preferred configuration is controlled by the sign of $W$, which favors particle-pairing for $W > 0$ but hole-pairing for $W < 0$. As a result, the state containing a particle dimer $|\psi_1\rangle$ is favored for $W > 0$, while $|\psi_2\rangle$ is preferred for negative $W$.

## 2.2  Duality and exotic dipole constraint

The behavior of the dipole TLL state is reminiscent of the so-called fractonic liquids that have been studied recently in the literature [14,30]. To clarify this link, we inspect more closely the point $W = J$ where the dipole-type operator $D$ becomes an exact symmetry of the lattice model.

The operator $D$ has been discussed before in the context of constrained quantum dynamics in one dimension [30,39]. In particular, Ref. [30] introduced $D$ as an ingredient to restrict spinless fermions to move in pairs and produce fractonic dynamics in one-dimensional polaronic systems. The hard-core boson ladder Eq. (1) has a similar behavior when $W = J$. As

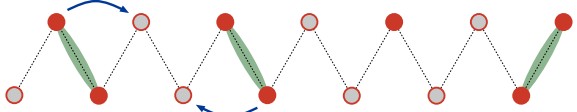

Figure 4: Cartoon of the strictly constrained point $W = J$. Dimers are depicted as bound pairs of bosons that move by *leapfrogging* their partners in the neighboring leg. Isolated particles are unable to move on their own.

illustrated in Fig. 4, bosons in one leg only move when assisted by a boson partner in the neighboring chain, binding them into a two-site molecule.

The nonlocal character of $D$ makes it hard to recognize its meaning. We thus move to a dual picture, by performing a Kramers-Wannier-like transformation [40], defined as

$$\tau_i^x = (-1)^i \prod_{j \leq i} \sigma_j^z, \qquad \tau_i^z = (-1)^i \sigma_i^x \sigma_{i+1}^x, \tag{6}$$

where we introduce oscillatory factors for convenience. Then, by replacing $n_i = \frac{1}{2}(1 + \sigma_i^z)$ into the formula for $\tilde{n}_i$ in Eq. (3), we learn $\tilde{n}_i$ is given by the difference $\tilde{n}_i = \frac{1}{2}(\tau_{i-1}^x - \tau_i^x)$. This implies that the dipole operator $D$ becomes the dual magnetization along the $x$ axis:

$$D = \frac{1}{2} \sum_i \tau_i^x, \tag{7}$$

where we assume an infinite system and drop boundary terms. Translating the hardcore boson model in Eq. (1) as well, we arrive at

$$H_{\text{KW}} = \sum_i \left[ W \tau_i^y \tau_{i+1}^y + J \tau_i^z \tau_{i+1}^z + \tau_{i-1}^x (J \tau_i^y \tau_{i+1}^y + W \tau_i^z \tau_{i+1}^z) \tau_{i+2}^x + V \tau_{i-1}^x \tau_{i+1}^x \right]. \tag{8}$$

The model $H_{\text{KW}}$ has a quite intriguing form. First, we readily recognize it commutes with $D$ when $W = J$, as $H_{\text{KW}}$ becomes manifestly invariant under rotations along the $x$ axis. The PXP-type constraint is perfectly implemented at the point $W = J$, where it becomes the folded XXZ model studied by Zadnik and Fagotti [23], who showed the existence of exponentially many jammed states. The $W = J$ model has also been considered by Yang et al. [20], who demonstrated that the Hamiltonian features Hilbert-space fragmentation and unusal thermalization properties even for nonzero $V$, away from the integrable point.

For $W \neq J$ the constraint is no longer perfectly implemented and the Hamiltonian $H_{\text{KW}}$ loses its invariance under U(1) rotations along the $x$-axis. The model still features a pair of U(1) conservation laws, inherited from the $N$ and $M$ numbers, given by the numbers of ferromagnetic and antiferromagnetic domain walls $\sum_i \tau_i^x \tau_{i+1}^x$ and $\sum_i (-1)^i \tau_i^x \tau_{i+1}^x$, which are conserved for all $W$. We note that Eq. (8) highlights the existence of a strong-weak $W$ coupling duality around $W = J$, implementing the exchange of $\tau^y \leftrightarrow \tau^z$ terms in the Hamiltonian.

## 3   Microscopic realizations of constrained dynamics

In this section we discuss potential realizations of our target Bariev-like model in atomic arrays. We use perturbation theory to examine two scenarios, one with Rydbergs in a linear chain and other where cold atoms move through the sites of an optical lattice. In both cases the central idea is the use of a strong potential bias to split the system into two sublattices, so that the slow, non-equilibrium dynamics preserves the relative number of excitations in each sublattice. Before continuing, let us note that some of the parameter regimes in which we are interested can also be achieved following earlier proposals, in particular, Refs. [34–36].

## 3.1 Rydberg-atom chain

We first illustrate a proposal utilizing Rydberg atoms trapped into optical potentials in the frozen regime, where atomic motion can be neglected [41]. We assume that atoms, prepared in two different Rydberg states, e.g. $|\circ\rangle = |nS\rangle$ and $|\bullet\rangle = |nP\rangle$, are placed in the sites of a linear chain. The dipolar coupling between Rydbergs produces a flip-flop (dipolar) exchange interaction $t_{ij}$ between pairs of atoms that decays approximately as $t_{ij} = t_{|i-j|} = t/|i-j|^3$. We also assume that the two atomic states are coupled by an external (microwave) drive. The Rydberg model Hamiltonian then reads

$$H_{\text{Ryd}} = \frac{\Omega}{2}\sum_i \sigma_i^x + \frac{\delta}{2}\sum_i (-1)^i \sigma_i^z + \sum_{i<j} t_{ij}(\sigma_i^+\sigma_j^- + \sigma_i^-\sigma_j^+), \qquad (9)$$

where $\Omega$ is the Rabi frequency, and $\delta$ is the staggered detuning corresponding to the drive. The staggering can be realized, e.g., by locally changing the Stark shift generated by the optical potential. To engineer the model in Eq. (1), we work in the regime where $\Omega \to 0$ and $\delta \gg t$. We take the zero Rabi frequency limit $\Omega \to 0$ so the total number of excitations is preserved, while the large $\delta$ limit allow us to freeze the magnetization $M$ and study hopping processes perturbatively.

**Perturbative treatment.—** We separate hopping terms in two groups. The first group does not change the number $M$, and comprises the hopping processes within the same sublattice. These tunneling amplitudes are hence not quenched by the detuning bias, popping out directly into the projected effective Hamiltonian. The second group, on the other hand, includes tunneling processes in which an excitation moves from one sublattice to the other. These processes are associated with a change $\Delta M = \pm 1$, and their leading contribution comes from second-order perturbation theory. Finally, given the rapid decay of the tunneling amplitudes, we truncate long-range hoppings beyond second-neighbors and make use of a Schrieffer-Wolff transformation (see Appendix A) to find

$$H_{\text{Ryd,eff}} = J\sum_i (\sigma_i^+\sigma_{i+2}^- + \text{H.c.}) - W\sum_i (-1)^i(\sigma_i^+\sigma_{i+1}^z\sigma_{i+2}^- + \text{H.c.}), \qquad (10)$$

where $J = t/8$, and $W = t^2/2\delta$. The effective model looks much like the Bariev chain [32], but the correlated exchange term acquires a staggering factor. The oscillatory phase is innocuous to our goal and favors pairing in the antisymmetric channel, giving rise to a dipole TLL made of particle-hole molecules. In fact the staggering phase in $W$ can be easily removed by performing a particle-hole transformation in just one of the legs, say

$$\mathcal{C}_1: \quad \sigma_{2l+1}^+ \leftrightarrow \sigma_{2l+1}^-, \qquad \sigma_{2l+1}^z \to -\sigma_{2l+1}^z. \qquad (11)$$

The resulting model, although with dominant antiferromagnetic correlations, is thus equivalent to Eq. (1) exhibiting a dipole TLL phase.

The Hamiltonian in Eq. (10) can only be taken as the dominant contribution in the more general case once longer-range terms are included. We however anticipate that those perturbations should not be harmful to the dipole TLL liquid. For instance, two perturbations that could arise are

$$\delta H_{\text{Ryd,eff}} \approx t_4 \sum_i \sigma_i^+\sigma_{i+4}^- + w_4 \sum_i (-1)^i \sigma_i^+(\sigma_{i+1}^z + \sigma_{i+3}^z)\sigma_{i+4}^- + \text{H.c.} + \cdots, \qquad (12)$$

where the estimated size of these couplings are $t_4 = t/64$ and $w_4 \sim t_1 t_3/\delta = t^2/27\delta$. These two terms break the integrability of the Bariev model, but are not enough to drive us away from the dipole TLL phase (as can be seen, at weak coupling, by analyzing their bosonized form). We thus argue Rydbergs are a promising platform to observe and study the dipole TLL state in quasi-adiabatic state preparations.

Figure 5: Illustration of the cold-atom ladder with second-order perturbative processes that arise in the quasi-adiabatic preparation. We assume $t'$ is small enough that we can neglect its corrections. The ladder has an artificial staggered flux configuration: triangles pointing up and down enclose zero and $\pi$ flux respectively.

## 3.2 Cold atoms with laser driven hopping

We now consider an alternative implementation, based on ultracold atoms trapped in the sites of an optical lattice [42]. We assume atoms are allowed to hop among first and second neighbors sites of a zigzag chain, and repel whenever two or more of them occupy the same site. The microscopic Hamiltonian then takes the form of a Bose-Hubbard (BH) model, with an additional chemical potential bias $\mu$ between even and odd sublattices:

$$H_{\text{BH}} = -t \sum_i (a_i^\dagger a_{i+1} + \text{H.c.}) - t' \sum_i (a_i^\dagger a_{i+2} + \text{H.c.}) + \frac{U}{2} \sum_i n_i^a (n_i^a - 1) + \frac{\mu}{2} \sum_i (-1)^i n_i^a, \quad (13)$$

where $a_i$ removes an atom sitting at $i$, $t$ and $t'$ are the hopping amplitudes, and $U$ is the onsite Coulomb repulsion. The number of atoms is denoted as $n_i^a = a_i^\dagger a_i$ to discern it from the hard-core boson number introduced before.

We utilize here laser-assisted tunneling [43–45] to induce hopping of atoms with a site-dependent phase. In particular, we consider a nearest-neighbor hopping $t \to t e^{i\varphi_{i,i+1}}$ with staggered flux pattern, depicted in Fig. 5. We choose the phase so its sole effect is to flip the sign of $t$ every two sites:

$$H_t \to H_t = -t \sum_l (-1)^l \left( a_{2l-1}^\dagger a_{2l} + a_{2l}^\dagger a_{2l+1} + \text{H.c.} \right), \quad (14)$$

where the $l$ sum is taken over half the system size. This phase dressing helps us to cancel out the oscillatory factor that arises in the perturbation theory, as we argue below.

**Perturbative treatment.—** The perturbative treatment is reminiscent of that in the previous subsection. We consider the large-$\mu$ limit to effectively enforce spin symmetry, and assume $U$ is strong enough so that every site contains at most one boson. It is worth noting that such strong couplings are not detrimental to a treatment in the single-band Hubbard regime, as the dynamics of single particles along the wires is not affected by those (in fact, similar regimes have been investigated in Ref. [46]).

We then take into account hopping processes perturbatively. As a further simplification, we consider the regime $t' \ll t$, so the second-neighbor hopping only contributes at first-order in perturbation theory (this assumption is not necessary, but makes computations easier to interpret). Taking into account second-order virtual processes generated by the first neighbor hopping, as seen in Fig. 5, we find the effective Hamiltonian governing the slow dynamics of the system takes the form

$$H_{\text{BH,eff}} = -\sum_{l\alpha} \left( J_\alpha \sigma_{2l-2+\alpha}^+ \sigma_{2l+\alpha}^- + W_\alpha \sigma_{2l-2+\alpha}^+ \sigma_{2l-1+\alpha}^z \sigma_{2l+\alpha}^- + \text{H.c.} \right) + V \sum_i \sigma_i^z \sigma_{i+1}^z, \quad (15)$$

where we use $\alpha = 1, 2$ to denote odd and even sub-lattices. The effective Hamiltonian features the desired $U(1)_c \times U(1)_s$ symmetry, but lacks leg permutation symmetry. From the Schrieffer-Wolff transformation, we estimate the couplings to be

$$J_{1,2} = t' \pm \frac{t^2}{U \pm \mu}, \qquad W_{1,2} = \frac{t^2}{\mu} \pm \frac{t^2}{U \pm \mu}, \qquad V = -\frac{t^2}{2}\Big(\frac{1}{U + \mu} + \frac{1}{U - \mu}\Big), \qquad (16)$$

where we take the upper sign for $\alpha = 1$ and the lower sign otherwise. There are two different ways to restore $\mathbb{Z}_2$ leg symmetry.[1] The first involves adjusting the relative filling in the legs, changing the free Fermi velocity so that they match at some nonzero magnetization $M \neq 0$. Another, less fine-tuned possibility, amounts to considering an extra separation of energy scales in the lattice parameters. Considering $U \gg \mu$ or $U \ll \mu$ will do the job. In the first case, for instance, by taking the limit $U \ll \mu$, with $t' \sim t^2/\mu$, we can approximate the couplings to $J_{1,2} \to J = t' + \frac{t^2}{\mu}$ and $W_{1,2} \to W = \frac{2t^2}{\mu}$, while $V$ goes to $V \to t^2 U/\mu^2$, and is then assumed to be much smaller than $J$ and $W$. In this limit, we thus arrive at precisely the model Hamiltonian in Eq. (1).

## 4   Effective field theory approach

In this section we present a long-distance, low-energy description for the hard-core boson ladder model. We use bosonization to analyze the effects of interactions starting from the limit of weakly coupled XY chains. As usual for TLLs, we can explore the predictions of the effective theory beyond the perturbative regime by treating the velocities and Luttinger parameters as phenomenological parameters. We shall see that this approach captures the transitions to the dipole TLL phase as well as to the classically ordered phases, providing a field theory framework for the entire phase diagram sketched in Fig. 3. In addition, we examine the representation of the dipole moment operator in the low-energy theory. We argue that the violation of the dipole symmetry is associated with the creation of gapped spin excitations that behave as mobile defects that interact with the gapless charge modes.

### 4.1   Tomonaga-Luttinger liquid theory

Let us write the hard-core boson ladder model in the limit of decoupled legs, obtained by setting $W = V = 0$ in Eq. (1). It is convenient to introduce a leg index $\alpha = 1, 2$ corresponding to odd and even sites, respectively, and denote the spin operators by $\sigma_\alpha^\pm(l) \equiv \sigma_{2l-2+\alpha}^\pm$, with $l \in \mathbb{Z}$. In this notation, the Hamiltonian for decoupled legs reads

$$H_0 = -J \sum_{l\alpha} [\sigma_\alpha^+(l)\sigma_\alpha^-(l+1) + \text{H.c.}]. \qquad (17)$$

We can then bosonize the low-energy excitations of each XY chain separately [40]. The effective Hamiltonian is that of two independent free bosons:

$$H_0 \approx \sum_\alpha \frac{v_0}{2} \int dx [(\partial_x \theta_\alpha)^2 + (\partial_x \phi_\alpha)^2], \qquad (18)$$

where $v_0 = 2J$ is the velocity of the bosonic modes in each leg and the bosonic fields obey the commutation relation $[\theta_{\alpha'}(x'), \partial_x \phi_\alpha(x)] = i\delta_{\alpha\alpha'}\delta(x - x')$. The fields $\phi_\alpha$ are associated with

---

[1]We note the lack of this $\mathbb{Z}_2$ symmetry is not necessarily incompatible with the dipole conservation. As a simple check one may consider the dipole-preserving, but still leg-anisotropic case where $W_1 = J_1 = J + \eta$ and $W_2 = J_2 = J - \eta$, which realizes an alternating hopping pattern for dimers, in a similar fashion to the Su-Schrieffer-Heeger model [47].

fluctuations of the hard-core boson occupation number by

$$\delta n_\alpha(l) = \frac{1}{2}\sigma_\alpha^z(l) \approx -\frac{1}{\sqrt{\pi}}\partial_x\phi_\alpha(x) + (-1)^x\text{const}\times\sin[\sqrt{4\pi}\phi_\alpha(x)]. \tag{19}$$

The staggered part has a nonuniversal prefactor and oscillates with momentum $\pi = \pi(1+\langle\sigma_i^z\rangle)$ for a pair of half-filled chains, described by $N/L \to 1/2$ and $M/L \to 0$. Note that if we allow the numbers $N$ and $M$ to change, the value of the momentum is not fixed and may be even different in each sublattice. The continuum expansion of the spin raising and lowering operators reads

$$\sigma_\alpha^\pm(x) \propto e^{\pm i\sqrt{\pi}\theta_\alpha(x)}\{1 + (-1)^x\text{const}\times\cos[\sqrt{4\pi}\phi_\alpha(x)]\}. \tag{20}$$

We note that the lattice model is invariant under discrete translations $i \to i+2$, corresponding to a rigid displacement of one site in each leg. In the low-energy theory, this lattice translation amounts to $x \mapsto x+1$ translations, under which the bosonic fields transform according to

$$\mathcal{L}: \qquad \phi_\alpha \mapsto \phi_\alpha + \frac{\sqrt{\pi}}{2}, \qquad \theta_\alpha \mapsto \theta_\alpha + \sqrt{\pi}. \tag{21}$$

In addition, the zigzag chain is invariant under a reflection about a site, which acts as site parity for the leg that contains that site, but link parity for the other leg. For instance, the reflection about an odd site acts in the low-energy theory as:

$$\mathcal{P}: \qquad x \mapsto -x, \quad \phi_1 \mapsto -\phi_1, \quad \phi_2 \mapsto \frac{\sqrt{\pi}}{2} - \phi_2, \quad \theta_1 \mapsto \theta_1, \quad \theta_2 \mapsto \theta_2 + \sqrt{\pi}. \tag{22}$$

We can also define time reversal as the anti-unitary transformation that takes $\boldsymbol{\sigma}_j \mapsto -\boldsymbol{\sigma}_j$. In the low-energy theory:

$$\mathcal{T}: \qquad i \mapsto -i, \quad \phi_\alpha \mapsto -\phi_\alpha, \quad \theta_\alpha \mapsto \theta_\alpha. \tag{23}$$

Next, we add interchain interactions perturbatively. We have $H = H_0 + H_W + H_V$, where

$$H_W = -W\sum_l\left[\sigma_1^+(l)\sigma_2^z(l)\sigma_1^-(l+1) + \sigma_2^+(l)\sigma_1^z(l+1)\sigma_2^-(l+1) + \text{H.c.}\right], \tag{24}$$

$$H_V = V\sum_l\sigma_2^z(l)[\sigma_1^z(l) + \sigma_1^z(l+1)]. \tag{25}$$

The three-spin interaction $H_W$ preserves $\mathcal{L}$ and $\mathcal{P}$ symmetries, but breaks $\mathcal{T}$. Using the continuum expansion of the spin operators, we can combine the oscillatory terms of both legs to produce the operator $\delta H_W \approx -\frac{2W}{\pi^2}\int dx\,\sin[\sqrt{4\pi}(\phi_1-\phi_2)]$ as the most relevant contribution. The Ising-like interchain interaction $H_V$ contributes with a marginal operator that couples the uniform magnetization in the two legs: $\delta H_V \approx \frac{8V}{\pi}\int dx\,\partial_x\phi_1\partial_x\phi_2$. We define the charge and spin fields as the linear combinations

$$\phi_{c,s} = \frac{\phi_1 \pm \phi_2}{\sqrt{2}}, \qquad \theta_{c,s} = \frac{\theta_1 \pm \theta_2}{\sqrt{2}}. \tag{26}$$

Adding the leading perturbations to Eq. (18), we obtain a spin-charge-separated Hamiltonian in the form $H = H_c + H_s$, where

$$H_c = \frac{v_c}{2}\int dx\left[K_c(\partial_x\theta_c)^2 + \frac{1}{K_c}(\partial_x\phi_c)^2\right],$$

$$H_s = \frac{v_s}{2}\int dx\left[K_s(\partial_x\theta_s)^2 + \frac{1}{K_s}(\partial_x\phi_s)^2\right] - \frac{\lambda}{2\pi^2}\int dx\,\sin(\sqrt{8\pi}\phi_s). \tag{27}$$

At weak coupling, the spin and charge velocities are given by $v_{c,s} \approx 2J(1 \pm 4V/\pi J)$. The Luttinger parameters $K_c$ and $K_s$ encode the interactions in the charge and spin sectors, respectively. To first order in the interleg interaction, we find $K_{c,s} \approx 1 \mp 4V/\pi$. The sine potential in the spin sector, with coupling constant $\lambda \approx 4W$, has scaling dimension $2K_s$. Note that this operator is odd under time reversal, as expected for the three-spin operator in $H_W$. Importantly, the low-energy Hamiltonian in Eq. (27) remains valid beyond the regime of small $W$ and $V$ because the sine potential is the leading perturbation compatible with $\mathcal{L}$, $\mathcal{P}$ and $U(1)_c \times U(1)_s$ symmetry.

The 2TLL phase corresponds to the regime in which both charge and spin sectors in Eq. (27) remain gapless. This can happen with the help of a repulsive interaction, $V > 0$, which disfavors the formation of pairs by making $K_s > 1$ and rendering the sine potential irrelevant. We can write the uniform part of the $\sigma_i^+$ operator in terms of charge and spin fields as

$$\sigma_{1,2}^+(l) \propto e^{i\sqrt{\pi/2}[\theta_c(x) \pm \theta_s(x)]}. \tag{28}$$

In the 2TLL phase, single-particle correlators display a power-law decay, given by

$$\langle 0 | \sigma_i^+ \sigma_{i+2r}^- | 0 \rangle \propto r^{-(K_c + K_s)/4K_c K_s}, \tag{29}$$

where $|0\rangle$ stands for the ground state. Note that in Eq. (29) we must take two points that belong to the same chain, otherwise the correlator vanishes identically. This is a consequence of the $U(1)_c \times U(1)_s$ global symmetry of the ladder and remains true even if we move away from the weak-coupling limit.

The transition to the dipole TLL phase is driven by the flow of the $\lambda$ perturbation to strong coupling. For $V = 0$, the coupling is marginally relevant, and, as a result, the spin sector undergoes a BKT-type transition. In the strong-coupling limit, we minimize the potential energy by pinning the scalar field $\phi_s$ to one of its minima:

$$\sqrt{8\pi}\phi_s \rightarrow \frac{\pi}{2} + 2\pi\mathbb{Z}. \tag{30}$$

The dipole TLL phase then corresponds to a gapless charge sector and a gapped spin sector. Note that attractive $V$ favors pairing, facilitating the transition to the dipole TLL state. Once the spin sector is gapped out, the single-particle propagator develops an exponential decay as follows:

$$\langle 0 | \sigma_i^+ \sigma_{i+2r}^- | 0 \rangle \propto e^{-r/\xi}/r^{1/4K_c}. \tag{31}$$

The correlation length $\xi$ is inversely proportional to the mass gap in the spin sector. In the case of a BKT transition at $V = 0$, the gap is exponentially small at weak coupling, with $\xi_{\text{BKT}}^{-1} \sim \exp(-\text{const}/\lambda)$ [40]. Power-law correlations in the dipole TLL phase are only found by pairing bosons in different sublattices, e.g.,

$$\langle 0 | \sigma_i^+ \sigma_{i+1}^+ \sigma_{i+r}^- \sigma_{i+r+1}^- | 0 \rangle \propto r^{-1/K_c}. \tag{32}$$

Note that the distance $r$ is not restricted to even multiples of the lattice spacing, cf. Eq. (29), since the two-particle operator $\sigma_i^+ \sigma_{i+1}^+$ creates one excitation in each leg, and the correlator always respects the $U(1)_c \times U(1)_s$ symmetry.

Instability towards phase separation is deduced from the vanishing of either charge or spin velocities in the large $V$ limit. Assuming a monotonic behavior and estimating the interaction dependence from the weak-coupling expressions for $v_c$ and $v_s$, we predict $V^\star/J \simeq \pm 0.78$, with positive and negative values corresponding to the transitions towards $PS_s$ and $PS_c$, respectively. Given that $W$ does not enter into the renormalization of Luttinger parameters at weak-coupling, we expect the critical value $V^\star$ to be roughly independent of $W$. We are thus led to the phase diagram shown in Fig. 3.

## 4.2 Emergent dipole symmetry

Let us now use our field theory formulation to examine the dipole operator in Eq. (3). Our goal here is to find its long-distance representation in order to verify it commutes with the low-energy Hamiltonian of the dipole TLL state, and thus represents an emergent symmetry of this phase.

We start from the dual representation in Eq. (7), where the dipole operator is given by the sum of Jordan-Wigner strings. We then expand each $\tau^x$ as the product of $\sigma^z$ strings in each leg. For an odd-site operator $\tau_1^x(l) = \tau_{2l-1}^x$, we get $\tau_1^x(l) = -\mathcal{S}_1(l)\mathcal{S}_2(l)$, where $\mathcal{S}_\alpha(l)$ is defined as

$$\mathcal{S}_1(l) = \prod_{m<l} \sigma_{2m}^z, \qquad \mathcal{S}_2(l) = \prod_{m<l} \sigma_{2m+1}^z. \tag{33}$$

Likewise, the even-site operator $\tau_2^x(l) = \tau_{2l}^x$ is given by $\tau_2^x(l) = \mathcal{S}_1(l)\mathcal{S}_2(l+1)$. We can then rewrite Eq. (7) as

$$D = -\frac{1}{2}\sum_l [\mathcal{S}_1(l) - \mathcal{S}_1(l+1)]\mathcal{S}_2(l), \tag{34}$$

where we sum over half the total number of sites.

We can now bosonize the dipole operator using the standard expression for the string operators in terms of the bosonic fields in each sublattice. Naive bosonization yields the complex form $\mathcal{S}_\alpha(l) \approx e^{i\frac{\pi}{2}x + i\sqrt{\pi}\phi_\alpha(x)}$. To obtain a manifestly Hermitian operator, we symmetrize the string, leading to

$$S_\alpha(l) \to S_{\alpha,\mathrm{reg}}(l) \approx \cos[\tfrac{\pi}{2}x + \sqrt{\pi}\phi_\alpha(x)]. \tag{35}$$

The product of strings at the same site gives $\mathcal{S}_1(l)\mathcal{S}_2(l) \approx \frac{1}{2}\cos[\sqrt{2\pi}\phi_s(x)]$, where we drop oscillatory terms and higher-order corrections. The second term in Eq. (34) is quite similar, but involves a derivative because the fields are at different points:

$$\mathcal{S}_1(l+1)\mathcal{S}_2(l) \approx -\frac{1}{2}\sin[\sqrt{2\pi}\phi_s(x)] - \frac{\sqrt{2\pi}}{4}\cos[\sqrt{2\pi}\phi_s(x)]\partial_x\phi_c(x) + \cdots \tag{36}$$

Taking both contributions into account, we arrive at the long-distance representation of the dipole operator:

$$D \propto \int dx \left[ \sin\left(\sqrt{2\pi}\phi_s + \tfrac{\pi}{4}\right) + \frac{\sqrt{\pi}}{2}\cos\left(\sqrt{2\pi}\phi_s\right)\partial_x\phi_c \right] + \cdots, \tag{37}$$

where we omit the prefactor and the ellipsis contains higher-order corrections.

From the continuum version of $D$, we readily learn that the dipole operator can only be a symmetry if $\phi_s$ condenses. This is clearly not the case in the 2TLL phase, where both $\phi_c$ and $\phi_s$ fluctuate and $D$ does not commute with the low-energy Hamiltonian in Eq. (27). In the dipole TLL phase, however, this condition is met as the spin field is pinned to $\sqrt{8\pi}\phi_s \to \pi/2$ according to the strong-coupling flow of $\lambda$. Plugging this condition into Eq. (37) gives

$$D \approx \frac{1}{\sqrt{2\pi}} \int dx \, \partial_x\phi_c + \cdots, \tag{38}$$

where we drop an unimportant additive constant, and fix the proportionality constant. In this form, the dipole operator is proportional to the total number of particles (half of it if we assume all particles are bound in pairs), and certainly commutes with the low-energy Hamiltonian.

This analysis tell us that the dipole operator describes an emergent symmetry of the dipole TLL phase. This symmetry is valid at low energies, below the spin gap, where all particles are confined into pairs but gapless excitations are still possible in the form of collective modes

in the charge sector. This finding supports the dimer picture of the dipole TLL phase. While our bosonization approach started in the weak-coupling limit where the spin gap is small, we expect this picture to become more accurate as we increase $W$ towards the special point $W = J$, where the dipole symmetry becomes exact and the pairs are strictly confined.

Single-particle excitations violate the pinning condition on $\phi_s$ because they carry both charge and spin quantum numbers. In the low-energy theory, the operator $\sigma_i^+$ in Eq. (28) creates a kink in the spin field at the position $x$, shifting $\phi_s$ as

$$\sqrt{8\pi}\phi_s(y) \to \sqrt{8\pi}\phi_s(y) + 2\pi\Theta_{\mathrm{H}}(y - x), \tag{39}$$

where $\Theta_{\mathrm{H}}(x)$ is the Heaviside step function. This means that the kink interpolates between two ground states of the sine potential in Eq. (27). On the other hand, the kink changes the sign of the dipole operator in Eq. (37) across the position where the single particle is created. This effect is consistent with the original definition of the dipole operator in Eq. (3), since inserting a single particle changes the sign of the string in $\tilde{n}_i$. Thus, we can view a local single-particle excitation as a defect in the spin field configuration. Far from this defect, we could still pin $\phi_s$ to a local minimum, but the dipole symmetry is spoiled if the defect is allowed to move through the lattice. In the following we will construct an effective mobile impurity model to describe the dynamics of this defect in the dipole TLL phase.

## 4.3 Mobile impurity model

According to Eq. (27), at low energies the spin sector of the dipole TLL is described by a sine-Gordon-type model. At weak coupling, i.e., for small spin gap $\Delta_s \ll J$, the elementary excitations are kinks or anti-kinks with relativistic dispersion $E_s(k) = \sqrt{v_s^2 k^2 + \Delta_s^2}$ [48]. As we increase the Bariev interaction strength $W$, the spin gap increases and the dispersion relation deviates from the relativistic dispersion. We are now interested in exploring the vicinity of the special point $W = J$, where the dipole symmetry imposes that single-particle excitations, which are charged under the $U(1)_s$ symmetry, cannot move by themselves. To describe this regime, we restrict the excitation spectrum to allow at most one spin excitation. This type of problem can be tackled using effective mobile impurity models, in which the finite-energy excitation is treated as a distinguishable particle that interacts with the gapless modes of the TLL [49].

We start by approximating the spin dispersion near its minimum by

$$E_s(k) \approx \Delta_s + \frac{k^2}{2m}, \tag{40}$$

where $m$ is the effective mass. For $W = J$, we expect $m \to \infty$, corresponding to localized spin excitations due to the exact dipole symmetry. We then treat the single spin excitation as an impurity mode, writing

$$\sigma_i^+ \propto e^{i\sqrt{\pi/2}\theta_c(x)} d_s^\dagger(x), \tag{41}$$

where $d_s^\dagger(x)$ is charge neutral but carries spin quantum number $\Delta M = +1/2$ ($-1/2$) if $i$ is an odd (even) site. In this representation, the ground state $|0\rangle = |0\rangle_c \otimes |0\rangle_s$ is a vacuum of the bosonic charge modes and of the $d_s$ particle. The effective mobile impurity model that includes the spin excitation has the form

$$H = H_c + \int dx \left[ d_s^\dagger \left( \Delta_s - \frac{1}{2m}\partial_x^2 \right) d_s - g\sqrt{\frac{2}{\pi}}\partial_x \phi_c d_s^\dagger d_s \right] + \dots, \tag{42}$$

where $H_c$ is the bosonic Hamiltonian in Eq. (27) and we omit irrelevant interaction terms. Phenomenologically, the coupling $g$ between the impurity and the charge density can be interpreted as follows. Suppose we perturb the hardcore boson ladder in the dipole TLL phase by

adding a uniform field $\delta H_Z = -\frac{h}{2}\sum_i \sigma_i^z$. In the low-energy theory, this perturbation becomes $\delta H_Z \approx h\sqrt{\frac{2}{\pi}}\int \mathrm{d}x\, \partial_x \phi_c$. In a grand canonical ensemble formulation, the change in the particle density can be absorbed by shifting the charge boson by $\phi_c(x) \to \phi_c(x) - \frac{hK_c}{v_c}\sqrt{\frac{2}{\pi}}x$. Implementing this shift generates a renormalization of the spin gap in the mobile impurity model in Eq. (42). As a result, we obtain the relation

$$g = \frac{\partial \Delta_s}{\partial \bar{\rho}_c}, \tag{43}$$

where $\bar{\rho}_c = \frac{1}{2}\langle \sigma_i^z + \sigma_{i+i}^z \rangle$ is the average charge density in the ground state and we use $\kappa = \partial \bar{\rho}_c / \partial h = 2K_c/\pi v_c$ for the charge compressibility of the TLL. Thus, the coupling constant $g$ depends on how the spin gap changes when we vary the total number of particles. This coupling is allowed when we have a finite-energy excitation in either spin or charge sectors [49–51].

We can eliminate the interaction between the impurity and the gapless modes using a unitary transformation. We define

$$U = \exp\left[ -i\sqrt{\frac{2}{\pi}}\frac{gK_c}{v_c}\int \mathrm{d}x\, \theta_c d_s^\dagger d_s \right] \tag{44}$$

and the transformed fields

$$\tilde{d}_s = U^\dagger d_s U = d_s e^{-i\sqrt{\frac{2}{\pi}}\frac{gK_c}{v_c}\theta_c}, \tag{45}$$

$$\partial_x \tilde{\phi}_c = U^\dagger \partial_x \phi_c U = \partial_x \phi_c - \sqrt{\frac{2}{\pi}}\frac{gK_c}{v_c}d_s^\dagger d_s. \tag{46}$$

In terms of the new fields, the Hamiltonian becomes

$$H = \int \mathrm{d}x\left[ \frac{v_c K_c}{2}(\partial_x \tilde{\theta}_c)^2 + \frac{v_c}{2K_c}(\partial_x \tilde{\phi}_c)^2 + \tilde{d}_s^\dagger\left( \Delta_s - \frac{1}{2m}\partial_x^2 \right)\tilde{d}_s \right] + \ldots, \tag{47}$$

where again we drop irrelevant interactions. Importantly, the dressed impurity mode $\tilde{d}_s$ carries a charge proportional to the coupling $g$ because the charge density operator is given by

$$\rho_c = -\sqrt{\frac{2}{\pi}}\partial_x \phi_c = -\sqrt{\frac{2}{\pi}}\partial_x \tilde{\phi}_c - \sqrt{\frac{2}{\pi}}\frac{gK_c}{v_c}\tilde{d}_s^\dagger \tilde{d}_s. \tag{48}$$

Since this impurity mode is non-interacting, we obtain the free propagator

$$G_d(x,\tau) = \langle 0|\tilde{d}_s(x,\tau)\tilde{d}_s^\dagger(0,0)|0\rangle = \int_{-\infty}^{\infty}\frac{\mathrm{d}k}{2\pi}e^{ikx - i\Delta_s \tau - ik^2\tau/2m - a^2 k^2/2}$$

$$= \frac{1}{\sqrt{2\pi}}\left( a^2 + \frac{i\tau}{m} \right)^{-1/2}\exp\left( -\frac{x^2}{2a^2 + 2i\tau/m} \right), \tag{49}$$

where $a^{-1}$ is a momentum cutoff, with $a$ of the order of the lattice spacing. Note that the propagator is a scaling function of $\tau/m$.

To probe the time evolution of the defect, let us consider the time-dependent variation in the local density:

$$C(j,\tau) = \langle \Omega| \sigma_0^- n_j(\tau)\sigma_0^+ |\Omega\rangle - \langle \Omega| n_j |\Omega\rangle, \tag{50}$$

where $|\Omega\rangle$ denotes the ground state of the hardcore boson ladder. In the mobile impurity model, this quantity becomes

$$C(x,\tau) \approx C_c(x,\tau) + C_s(x,\tau). \tag{51}$$

The first term involves the charge density. Defining the vertex operator $V_c = e^{-i\sqrt{\pi/2}(1-g\kappa)\tilde{\theta}_c}$, we obtain $C_c(x,\tau) \propto \langle 0|V_c(0)\partial_x\phi_c(x,\tau)V_c^\dagger(0)|0\rangle_c$. This contribution spreads ballistically with the charge velocity $v_c$ and decays algebraically at long times. The second contribution involves the impurity propagator:

$$C_s(x,\tau) \propto \langle 0|\tilde{d}_s(0)\tilde{d}_s^\dagger(x,\tau)\tilde{d}_s(x,\tau)\tilde{d}_s^\dagger(0)|0\rangle_s = G_d(x,\tau)G_d(-x,-\tau). \tag{52}$$

Using Eq. (49) and setting $x = 0$, we find that the density measured at the same position where the defect is created decays with time as $C(0,\tau) \propto m/\tau$ for any finite mass. In the limit of an immobile defect, $m \to \infty$, $C(0,\tau)$ converges to a finite non-universal value for $\tau \to \infty$, implying that some charge remains at $x = 0$ while the other fraction propagates away with velocity $v_c$.

We can also use the mobile impurity model to study the dynamics of the dipole moment within the low-energy theory. To reproduce the properties of the dipole operator in Eqs. (3) and (38), we propose the following expression in the continuum:

$$D \approx \int_{-\infty}^{\infty} dx\, x d_s^\dagger(x)d_s(x) + \frac{1}{\sqrt{2\pi}}\int_{-\infty}^{\infty} dx\, \hat{S}(x)\partial_x\phi_c(x), \tag{53}$$

where we define $\hat{S}(x) = 1 - 2\int_{-\infty}^{x} dx'\, d_s^\dagger(x')d_s(x')$. The first term simply accounts for the dipole moment of the single defect. The second term represents the contribution from bound pairs, where $\hat{S}(x)$ implements the sign change of the string when we cross the position of the defect. In the defect-free sector, the impurity density vanishes identically, and the dipole operator reduces to the total number of pairs, a conserved quantity within the low-energy theory. By contrast, if we consider the initial state $|\Psi(\tau = 0)\rangle = \sigma_j^+ |\Omega\rangle$, we expect the variance of the dipole operator to increase with time as the defect moves through the system. We can capture this effect by calculating the variance due to the first term in Eq. (53). We obtain

$$\langle\Delta D^2(\tau)\rangle = \langle\Psi(\tau)|D^2|\Psi(\tau)\rangle - \langle\Psi(\tau)|D|\Psi(\tau)\rangle^2 \approx \int dx\, x^2|G_d(x,\tau)|^2, \tag{54}$$

which can be interpreted as the mean squared displacement of the defect. We find

$$\langle\Delta D^2(\tau)\rangle \approx a^2 + \left(\frac{\tau}{ma}\right)^2. \tag{55}$$

The variance is finite at $\tau = 0$ because the initial state is not an eigenstate of the dipole operator. For any finite mass, the variance is a function of $\tau/m$ and increases quadratically with time. For $m \to \infty$, the variance remains approximately constant, in agreement with the picture of a localized defect enforced by the exact dipole moment conservation law.

# 5  Numerical simulations

In this section we present our numerical findings, obtained from a combination of ED and matrix product states (MPS) based methods, such as TDVP and TEBD. We begin with an examination of the phase diagram from the viewpoint of the emergent dipole conservation. We use ED results to provide insight into the conservation of the dipole moment in both ground state and low-energy excited states. We then move to the benchmark of the quasi-adiabatic state preparation. We focus on the more challenging preparation with cold atoms, simulating the dynamical preparation with TDVP. We close our numerical survey by studying the isolated defect dynamics close to the dipole TLL ground state.

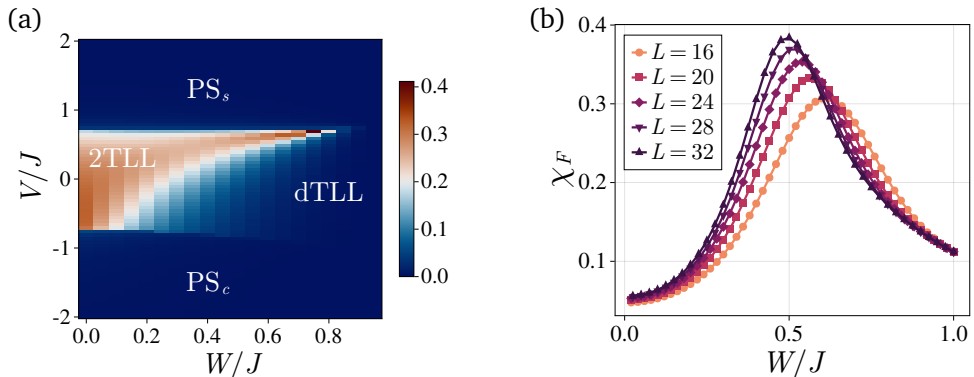

Figure 6: Ground state characterization with ED. (a) Phase diagram showing the variance of the dipole operator $\langle \delta D^2 \rangle / L$ as a function of $W$ and $V$ for an open chain with $L = 28$ sites. (b) Fidelity susceptibility at $V = 0$ for various system sizes with periodic boundary conditions.

## 5.1 Phase diagram and emergent dipole symmetry

We start with a phase diagram characterization of the dipole symmetry. Our goal here is not to precisely determine transition points, but rather to uncover the emergent status of the dipole symmetry in the dipole TLL state.

We first check how the ground state variance of the dipole operator, $\langle \Delta D^2 \rangle = \langle D^2 \rangle - \langle D \rangle^2$, behaves as we navigate across different portions of the phase diagram. In Fig. 6(a) we plot the results obtained for a chain with $L = 28$ sites and open boundary conditions. We observe a great similarity with the phase diagram sketched in Fig. 3. Near the origin, where we find the 2TLL state, we see the ground state is far from being dipole symmetric as flagged by the higher variance. Moving either up or down we eventually cross to vanishing dipole variance regions, which we associate to the classically ordered states. Coincidentally, we find the transitions take place around the values $V/J \approx \pm 0.75$, quite close to the ones predicted from the weak-coupling bosonization. Finally, when we leave the 2TLL state moving in the direction of increasing $W$, we appear to cross a smoother region after which the dipole variance also approaches zero. This would correspond to the BKT-type transition into the dipole TLL state, supporting the picture of an emergent dipole-conserving liquid groundstate.

The BKT nature of the transition to the dipole TLL phase makes it difficult to use finite-size scaling techniques effectively. This limitation leads to an overestimation of the 2TLL phase in finite-size numerics, as can be seen in Fig. 6(a). This becomes clear when we compute the fidelity susceptibility $\chi_F$, conventionally used to detect critical points via finite-size scaling techniques [52] and defined as

$$\chi_F(\eta) = \lim_{\delta\eta \to 0} \frac{2}{L} \frac{1 - F(\eta, \delta\eta)}{(\delta\eta)^2}, \tag{56}$$

where $F(\eta, \delta\eta) = |\langle \psi(\eta) | \psi(\eta + \delta\eta) \rangle|$ is the fidelity and $\eta$ is a parameter of the Hamiltonian. We show in Fig. 6(b) the ground state fidelity susceptibility as a function of $W/J$ for a few different sizes $L$. We find that the transition slowly moves towards weak coupling as we increase the size $L$, indicating that the 2TLL may actually be much smaller in the thermodynamic limit.

Next, we address the behavior of the dipole operator for excited states. Figure 7 shows the matrix elements $|D_{\alpha\beta}| = |\langle \alpha | D | \beta \rangle|$ of the dipole operator, computed in the basis of eigenstates for $W/J = 0, 0.60$ and $0.90$. Note that we plot absolute values, so we can focus on the strength of nonzero terms. We observe off-diagonal terms are gradually suppressed as we move from $W/J = 0$ to $W/J = 0.90$, in support to the emergent status of the symmetry. In particular,

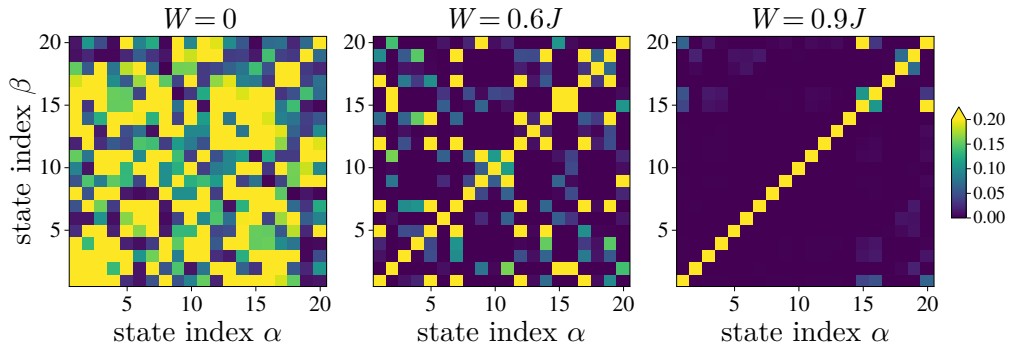

Figure 7: Matrix elements of the dipole operator for three values of $W$. We plot the absolute values $|D_{\alpha\beta}| = |\langle\alpha|D|\beta\rangle|$, obtained from the 20 lowest-energy excited states. Data obtained from the ED of an open chain with $L = 28$ sites and $V = 0$.

for $W/J = 0.90$, we can see a low-energy block, whose nonzero elements are concentrated on the diagonal. For $W = J$ (not shown), the lattice model commutes with $D$, ensuring that energy eigenstates possess a definite dipole number: $D|\alpha\rangle = d_\alpha|\alpha\rangle$. Consequently, the matrix elements of $D$ become diagonal in the energy eigenstate basis $D_{\alpha\beta} = d_\alpha\delta_{\alpha\beta}$. Notably, introducing a nonzero $V$ does not alter these results, as it preserves the dipole symmetry.

## 5.2 State preparation

Let us now consider the state preparation protocol. We start from the simple initial state $|\text{Néel} \times \text{Néel}\rangle = |01100110\ldots\rangle$, and evolve it according to the following time-dependent Hamiltonian:

$$H_{\text{exp}}(\tau) = H_{\text{BH}}(\tau) + h(\tau)\sum_l (-1)^l(n^a_{2l} + n^a_{2l+1}), \tag{57}$$

where $H_{\text{BH}}(\tau)$ is a time-dependent variant of the Bose-Hubbard model shown in Eq. (13). At the initial time, $\tau = 0$, coupling parameters are chosen so the initial state is the actual ground state of the full Hamiltonian $H_{\text{exp}}(0)$ in the symmetry sector where both sublattices are half-filled. This means, initially, the Bose-Hubbard model only includes the potential terms $U = U_0$ and $\mu = \mu_0$, while the hopping elements are set to zero, $t_0 = t'_0 = 0$. Note that we also add an extra time-dependent staggering field $h(\tau)$, whose initial value $h = h_0$ favors the $|\text{Néel} \times \text{Néel}\rangle$ configuration.

The preparation then proceeds by slowly tuning the coupling parameters in $H_{\text{exp}}(\tau)$. We vary three parameters $t$, $t'$, and $h$, while keeping the potentials $U$ and $\mu$ static along the evolution. The hopping parameters $t$ and $t'$ are increased up to the terminal values $t_f = 1$ and $t'_f = 0.1$, while $h$ is decreased all the way down to $h_f = 0$. The parameter sweep is shown in Fig. 8(a) as a function of $\tau/T$, where $T$ denotes the duration of the sweep. We use ED to verify how the many-body spectrum evolves along our parameter flow. In Fig. 8(b) we plot the spectrum evolution of the Hamiltonian (57) for a modest chain with $L = 16$ sites. We observe the low-energy manifold remains separated, below the rest of the spectrum, during the whole evolution. Note that we choose the parameters to be as close as possible to the strictly confined point of the Bariev-like model, so we perform the ED in the limit of hard-core bosons ($U \to \infty$) with potential bias set to $\mu = 10$. In this parameter regime, we should ideally end up in the effective model of Eq. (15), with parameters $J_\alpha = t'$, $W_\alpha = t^2/\mu$, and $V = 0$, as estimated from Eq. (16).

We now consider a quasi-adiabatic protocol with a finite preparation time $T$ [37]. To perform the dynamical evolution of the initial state we then resort to TDVP, using a MPS representation of the boson states with maximum occupation number equal to four. We study this

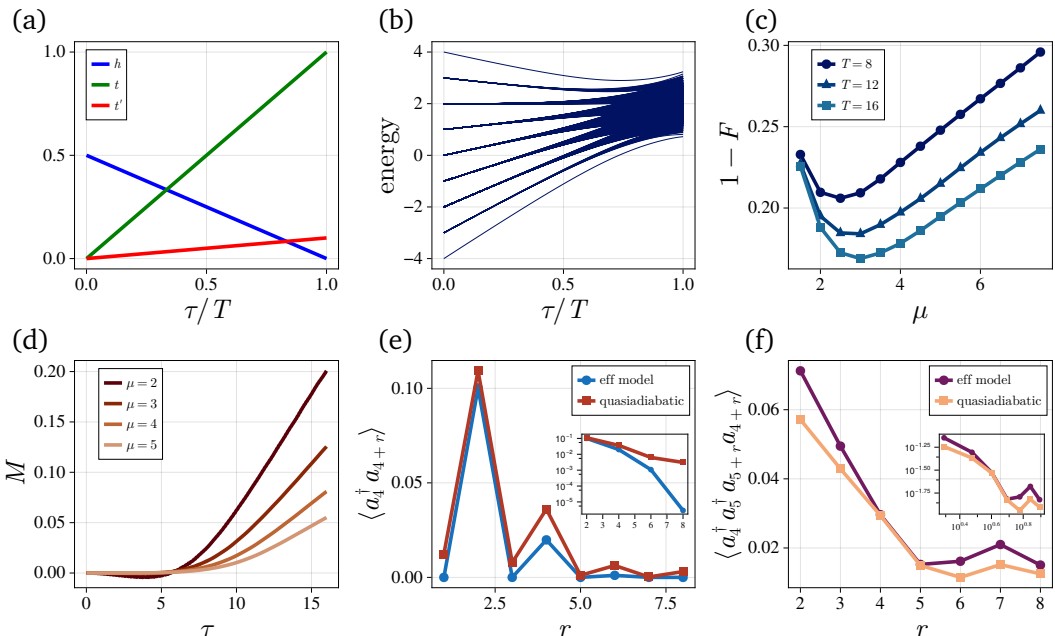

Figure 8: Quasiadiabatic preparation protocol. (a) Parameter sweep profile employed. (b) Energy spectrum evolution along the parameter sweep. Obtained from the ED of $H_{\text{exp}}(\tau)$ in the limit of hardcore bosons, with $\mu = 10$ and a chain with $L = 16$ sites. (c) Overlap between the dynamically prepared state in the TDVP evolution and the ground state of the target model as a function of $\mu$. (d) Time evolution of the inter-leg magnetization for the initial $|\text{Néel} \times \text{Néel}\rangle$ state. (e) Single-particle and (f) two-particle correlators of the dynamically prepared state, compared with the correlators obtained from the ED of the target model. Insets in (e) and (f) are respectively the log-linear and log-log behavior of the corresponding correlators. For the inset in (e) we only plot even values of $r$.

preparation as a function of the duration time $T$ and the potential bias $\mu$, fixing the on-site potential to $U = 50$.

Figure 8(c) shows the behavior of the overlap between the dynamically prepared and the (DMRG obtained) target state of the Bariev model, $F = |\langle \Psi_{\text{prep}}(T)|\Psi_{\text{target}}\rangle|$. As expected, we see that the fidelity improves with increasing $T$ (or, equivalently, with decreasing sweep rate). We also observe that the maximal overlap is reached at intermediate $\mu$, while small and large values of $\mu$ lead to significantly smaller overlaps with the target state. For small $\mu$, higher-order terms in the perturbation theory become sizeable and cannot be neglected. On the other hand, for large $\mu$, the energy gaps, which scale as $t^2/\mu$, become small, such that a higher density of excitations are created in the quasi-adiabatic protocol. We then verify that, while the experimentally prepared state does not conserve the magnetization number $M$ exactly, its expectation value approaches zero with increasing $\mu$ as shown in Fig. 8(d).

To conclude, we compare the behavior of the single- and two-particle correlators obtained at the end of the protocol with the ones computed from the ED of the target, hardcore boson model Eq. (1), with $W = J$ and $V = 0$. This test is important to address the question, whether qualitative features of the model ground state are observable in experiments, without necessarily comparing quantiative aspects such as the precise decay of correlations. As shown in panels (e) and (f) of Fig. 8, these are in good agreement with the ideal results obtained from the target model. We point out, however, that the single-particle correlator exhibits greater deviations at larger distances, aligning with predictions from the Kibble-Zurek mechanism [53, 54]. We have also tried the state preparation protocol in the limit $U \ll \mu$. However, we

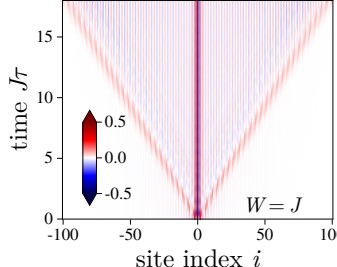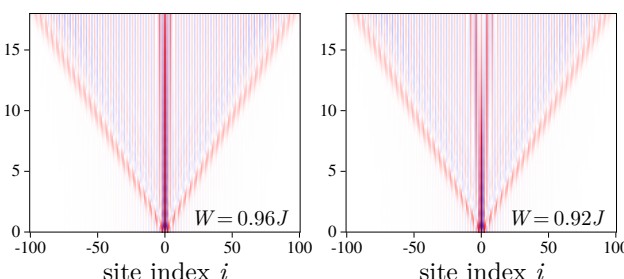

Figure 9: Defect dynamics on top the dipole TLL state. Time evolution of the local density variation $\langle\delta n_i(\tau)\rangle$ for three different values of $W$. From left to right, $W/J = 1$, 0.96, and 0.92. The Ising-like interaction is set to $V/J = 0.1$ in all cases. Results obtained with TEBD on a chain with $L = 201$ sites.

observe that the state does not appear to enter the dipole TLL phase, using experimentally realizable parameter regimes.

## 5.3 Dynamics of defects

Finally, we investigate the out-of-equilibrium dynamics of single-particle excitations. The philosophy here is that, after approaching a target ground state in the quasi-adiabatic preparation, one acts locally with an operator that creates an excitation in the center of the chain letting it evolve coherently for some time. With this in mind, we however leave microscopic models behind and concentrate on the ground state and dynamics produced by the effective hardcore boson ladder.

Numerically our quench protocol goes as follows. First, we use DMRG to prepare the ground state $|\Omega\rangle$ of the Bariev-like Hamiltonian, Eq. (1). We then act with $\sigma_0^+$, where $j = 0$ represents the center site of a chain with an odd number of sites. Finally, we use a three-site gate TEBD to approximate the time evolution $|\Psi(\tau)\rangle = e^{-iH\tau}|\Psi_0\rangle$, with $|\Psi_0\rangle = \sigma_0^+|\Omega\rangle$ the prepared initial state.

We apply this recipe to examine how the defect behaves above the dipole TLL ground state. In order to get cleaner results, we choose lattice parameters so we are deep in the dipole TLL phase, close to the strictly confined point $W = J$. We consider values of $W$ in the range from $W/J = 1$ to $W/J = 0.92$, always with a small $V/J = 0.1$ interaction for generality. We run the DMRG for a chain with $L = 201$ sites and quantum numbers fixed to $N = (L-1)/2 = 100$ and $M = 0$. For the TEBD part, we use moderately small time steps $J\delta\tau = 0.01$, and stop the evolution at $J\tau = 18.0$, roughly when the light cone reaches the edges of the chain. This is done in order to mitigate finite-size effects. Indeed, in the field theory description, given the characteristic velocity $v_c$ of charge excitations, the distance $d$ to the edges sets a time scale $T \sim d/v_c$. Beyond this time scale, finite-size effects become appreciable. The maximum truncation error is set to $10^{-8}$ during the whole numerical experiment. Attained with a maximum bond dimension of $\chi = 1000$, the results appear well-converged for the times considered.

In Fig. 9 we plot the time evolution of the variation in the local density for three different values of $W$. In agreement with the field theory prediction, we observe that the defect exhibits two different spreading patterns as it carries both quantum numbers of charge and spin. After emitting the gapless part, which spreads ballistically and leaves a clear light-cone signal, the remaining part of the defect features a substantial slowdown in the relaxation towards equilibrium. For $W/J = 1$ the effect is most dramatic, since the spin part of the defect still lies at the central site where it was created. However, as we move away from the strictly confined point, we are able to observe a slow spreading of the contribution associated with the defect.

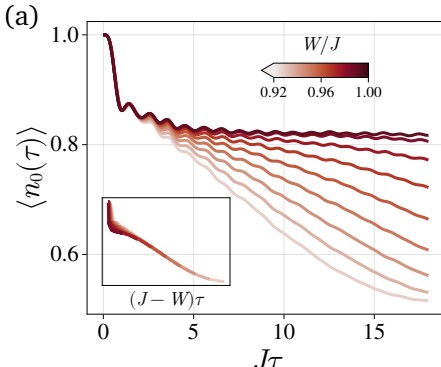 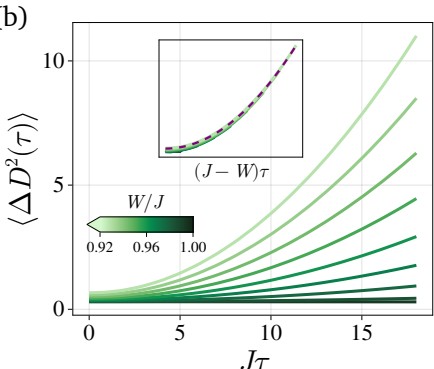

Figure 10: Time correlators for the defect in the dipole TLL state. (a) Time evolution of the charge density in the central site of the chain for different values of $W$ ranging from $W/J = 1$ (darker) to $W/J = 0.92$ (lighter). The inset shows the rescaling of the time axis by the factor $J - W$. (b) Time evolution of variance of the lattice dipole operator. Inset shows the rescaling of the time axis by the factor $J - W$. The purple dashed line represents a quadratic behavior of the form $a + b\tau^2$, with $a$ and $b$ constants. Results obtained via TEBD on a chain with $L = 201$ sites.

We stress that the field theory should be applicable for the quench dynamics as long as the latter is dictated by low-energy excitations [55, 56]. As the action of $\sigma_j^+$ also creates a defect above the gap, we include its effect through the picture of heavy mobile impurities.

The time evolution of the density at the central site for various values of $W$ is shown in Fig. 10(a). There we can spot two time regimes. At short times, we observe a rapid decay due to the formation of the wave front that spreads ballistically. After this initial decay, the density leak slows down. In particular, for $W = J$ the density does not appear to decay significantly for the time scales observed, staying close to $\langle n_0 \rangle \approx 0.8$. This is consistent with the limit $m \to \infty$ of the mobile impurity model, where a finite amount of the density excess remains localized at infinitely long times. For $W < J$ we start to observe a slow decay in the density. The inset of Fig. 10(a) we show that these curves collapse onto a single curve upon scaling time has been rescaled by $(J - W)\tau$. The data collapse gives the estimate $m \sim |W - J|^{-1}$ for the behavior of the effective mass parameter in the mobile impurity model. Note however in the available time windows we could not directly verify the long-time behavior of $1/\tau$ predicted by the impurity model, controlled by the limit $\tau \gg ma^2$, with $a$ a short-distance cutoff.

We show the time evolution of the variance of the dipole operator in Fig. 10(b). As expected, we observe that the dipole variance vanishes for $W = J$ and increases with the difference $|W - J|$. We use this correlator to cross-compare time scales as extracted from the time evolution of the central site density. As shown in the inset of Fig. 10(b), the time scales here are also compatible with an effective mass for the mobile impurity model given by $m \sim 1/|W - J|$, displaying a quadratic form as predicted in Eq. (55).

To complete the picture, we consider the same quench protocol but with a defect added on top of the 2TLL state. However, due to the large entanglement pattern of the $c = 2$ state, we observe a greater difficulty to prepare well-converged ground states with DMRG. In view of that we reduce the system size and consider a modest chain of $L = 49$ sites. We set the lattice parameters to $W = 0$ and $V/J = 0.1$, setting the truncation error to $10^{-12}$, with maximum bond dimension to $\chi = 600$ during the ground state search. The rest of the numerical protocol goes the same as before and we arrive at the results shown in Fig. 11.

Given that the 2TLL phase is adiabatically connected to the fixed point of decoupled chains, the numerical results confirm the natural expectation, and demonstrate the defect thermalizes quickly, spreading ballistically throughout the system. In Fig. 11(a) we plot the spacetime de-

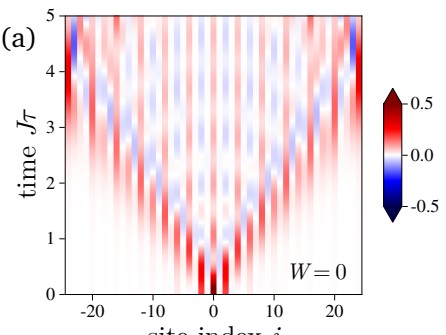 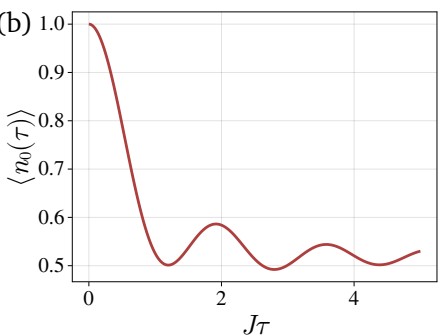

Figure 11: Ballistic dynamics for defect in the 2TLL phase. (a) Spacetime dependence of the charge excess. (b) Time evolution of the charge density at the central site. Here the parameters of the lattice model are $W = 0$, and $V/J = 0.1$. Results obtained by TEBD on a chain with $L = 49$ sites.

pendence of the charge variation $\langle \delta n_j(\tau) \rangle$ during the quench with respect to the unperturbed ground state. Note that for the system size considered, and small difference in velocities, it is difficult to distinguish the light cones associated with the fractionalization of the single-particle excitation into gapless modes of charge and spin. Another contrast is provided in Fig. 11(b), where we plot the time evolution for the number occupation at the central site. When compared to the behavior in the dipole TLL phase, we can see the charge leak does not exhibit any sort of slowdown, relaxing at time scales of the order $J\tau \sim 1$. In appendix B we include further results obtained for the pair dynamics on top the dipole TLL state.

# 6 Outlook

We investigated a Bariev-type model for hard-core bosons and its non-equilibrium implementation in biased atomic ladders. We considered how deviations from a strictly confined point still leave significant imprints on low-energy physics. In particular, we uncovered the emergence of a global nonlocal symmetry in the dipole TLL state, which constrains the dynamics in the ground state, binding excitations into pairs. Through the use of extensive numerical methods, we simulated and verified the effectiveness of the quasi-adiabatic preparation in the context of a Bose-Hubbard model. We further considered the out-of-equilibrium dynamics of single-particle defects created above the dipole TLL state. We showed that they exhibit a substantial slowdown in the spreading dynamics and compared their slow motion to that of a heavy particle whose mass diverges as we approach the special point where the dipole symmetry is exact.

We leave some open directions for future work. The continuum-limit description of single-particle defects may be improved by treating the sine-Gordon model in its entirety, which may open the possibility of better understanding the role of the emergent constraint as well as extending the theory to finite densities of such defects. Another promising direction is the exploration of a potential link to the theory of $\mathbb{Z}_2$ lattice gauge theories coupled to spinless fermions [26, 28], which exhibits a Bariev-type dynamics in the strong-string-tension limit. Drawing inspiration from works on the folded XXZ model [23, 24], it may also be of interest to explore the role of the emergent dipole symmetry (if any) in the realm of finite-temperature transport [7].

Our predictions are of particular relevance to atomic physics experiments, where the protocol we describe could be implemented in a controllable way. We note that the use of staggering

fields has also been proposed to implement gauge symmetries [57] in quantum analog simulators, as an alternative to tilting potentials [2–4] which need to scale with system size. The core idea of splitting a bipartite lattice with a bias potential is quite generic, and one may envision extensions to 2D systems, where the number of excitations is separately conserved in each sublattice. These ideas might also be of relevance to trapped ion chains, where the implementation of three-body terms similar to those discussed here has recently been proposed [58], or via effective dynamics similar to the Rydberg case, but at the price of introducing longer-range interactions [59, 60].

# Acknowledgements

We thank Devendra Bhakuni, Michael Knap, Sergej Moroz and Lenart Zadnik for insightful discussions and pointing out key references. H.B.X. expresses his gratitude to Hosho Katsura for stimulating discussions about Bariev-type models and their cold-atom realizations. M.D. thanks Guido Pagano for comments on the manuscript, and enlightening discussions about trapped ion systems. Matrix product state simulations were performed using the ITensor package [61, 62].

**Funding information** M. D. was partly supported by the QUANTERA DYNAMITE PCI2022-132919, by the EU-Flagship programme Pasquans2, by the PNRR MUR project PE0000023-NQSTI, and by the PRIN programme (project CoQuS). H. B. X. and M. D. were partly supported by the MIUR Programme FARE (MEPH). P. S. T. acknowledges support from the Simons Foundation through Award 284558FY19 to the ICTP. R. G. P. acknowledges support from the Conselho Nacional de Desenvolvimento Científico e Tecnológico and by a grant from the Simons Foundation (Grant No. 1023171).

# A   Details on the Schrieffer-Wolff transformation

Here we offer further details on the Schrieffer-Wolff transformation [63, 64] we employ to derive the effective Hamiltonians for the Rydberg and cold atom platforms.

## A.1   Rydbergs

The Rydberg Hamiltonian has the form $H_{\text{Ryd}} = 2\delta M + H_t$, where $M = \frac{1}{4}\sum_i (-1)^i \sigma_i^z$, and $H_t = \sum_{ij} t_{ij} \sigma_i^+ \sigma_j^-$ describes long-range tunneling. Given that amplitudes decay quite fast $t_{ij} = t_{|i-j|} = t/|i-j|^3$, we truncate the hopping up to second neighbors. The projection of $H_{\text{Ryd}}$ into a fixed sector of $M$, say $M = 0$, then gives

$$H_{\text{Ryd,eff}} = P_0 H_t P_0 = t_2 \sum_i (\sigma_i^+ \sigma_{i+2}^- + \sigma_i^- \sigma_{i+2}^+), \tag{A.1}$$

where $t_2 = t/8$, and $P_0$ is the projector onto the $M = 0$ subspace. Second-order corrections are obtained from the Schrieffer-Wolff transformation $H \to H' = e^S H e^{-S}$. Treating only the first-neighbor hopping $t_1 = t$, we choose the generator so that $[S, M] = -V/2\delta$, where $V$ is the off-diagonal element that connects the $M = 0$ manifold to the subspaces labeled with $M = \pm 1$. With the help of projectors onto the relevant manifolds, $P_\pm \equiv P_{\pm 1}$, we write $V$ as

$$V = P_0 H_t P_+ + P_+ H_t P_0 + P_0 H_t P_- + P_- H_t P_0. \tag{A.2}$$

From it we find the suitable choice for $S$ reads

$$S = \frac{1}{2\delta}(-P_0 H_t P_+ + P_+ H_t P_0 + P_0 H_t P_- - P_- H_t P_0). \tag{A.3}$$

The second-order term comes from the commutator $[S, V]$, namely $\delta H = \frac{1}{2} P_0 [S, V] P_0$. From the projection operator properties, we can simplify this commutator to

$$\delta H_{\text{Ryd,eff}} = -\frac{1}{2\delta}(P_0 H_t P_+ H_t P_0 - P_0 H_t P_- H_t P_0). \tag{A.4}$$

We observe that the two signs are different, because excitations increase energy when go from odd to even sublattices, but lower energy when move in the opposite direction. Writing out the explicit formula for the hopping, we fix initial and final sites with the use of projection operators, and the expression becomes

$$\delta H_{\text{Ryd,eff}} = -\frac{t^2}{2\delta} \sum_{ij} \left[ (\sigma_{2i-1}^+ + \sigma_{2i+1}^+) \sigma_{2i}^-, \sigma_{2j}^+ (\sigma_{2j-1}^- + \sigma_{2j+1}^-) \right]. \tag{A.5}$$

Applying the commutator identity $[AB, CD] = A[B, C]D + AC[B, D] + [A, C]DB + C[A, D]B$, we then arrive at

$$\delta H_{\text{Ryd,eff}} = -\frac{t^2}{2\delta} \sum_i (-1)^i (\sigma_i^+ \sigma_{i+1}^z \sigma_{i+2}^- + \text{H.c.}). \tag{A.6}$$

We draw attention to the fact that Ising terms cancel out due to the staggering behavior of second-order processes. Together with Eq. (A.1) this leads us to the formula presented in the main text.

## A.2   Cold atoms

We write the Bose-Hubbard model, Eq. (13), as $H_{\text{BH}} = UC + \mu M_a + H_t + H_{t'}$. The first two terms are dominant. They represent the on-site Coulomb repulsion, $C = \frac{1}{2} \sum_i n_i^a (n_i^a - 1)$, and the staggered potential, $M_a = \frac{1}{2} \sum_i (-1)^i n_i^a$. We factor the couplings out to ease the power counting. The other two terms describe first- and second-neighbor site hopping with tunneling amplitudes $t$ and $t'$. Projection onto the subspace manifold where all sites are either empty or singly-occupied, and there is an equal number of atoms in the two sublattices gives

$$H_{\text{BH,eff}} = P_{s0} H_{\text{BH}} P_{s0} = -t' \sum_i (\sigma_i^+ \sigma_{i+2}^- + \sigma_i^- \sigma_{i+2}^+), \tag{A.7}$$

where $\sigma_i^- = P_{s0} a_i P_{s0}$, with $P_{s0}$ the projector onto the $C = M_a = 0$ subspace. We now assume $t' \ll t$, so at second-order we only consider the effect of $H_t$. The off-diagonal elements generated by the $H_t$ in Eq. (14) can be organized as

$$\begin{aligned} V = &P_{s0} H_t P_{s+} + P_{s+} H_t P_{s0} + P_{s0} H_t P_{s-} + P_{s-} H_t P_{s0} \\ &+ P_{s0} H_t P_{d+} + P_{d+} H_t P_{s0} + P_{s0} H_t P_{d-} + P_{d-} H_t P_{s0}. \end{aligned} \tag{A.8}$$

The first line describes hopping processes where the atom hops from one sublattice to the other landing on an empty site. This matrix element involves a change $\Delta M_a = \pm 1$ but not on $\Delta C = 0$, as indicated by the projectors $P_{s\pm}$. In contrast the second line represents tunneling processes where the atom arrives at a site that is already occupied. Hence both quantum numbers change: $\Delta C = 1$ and $\Delta M_a = \pm 1$, so we use $P_{d\pm}$ to indicate we are within the

manifold with one doubly-occupied site. To perform the Schrieffer-Wolff transformation, we pick the generator

$$
S = \frac{1}{\mu}(-P_{s0}H_t P_{s+} + P_{s+}H_t P_{s0} + P_{s0}H_t P_{s-} - P_{s-}H_t P_{s0})
$$

$$
+ \frac{1}{U+\mu}(-P_{s0}H_t P_{d+} + P_{d+}H_t P_{s0}) + \frac{1}{U-\mu}(-P_{s0}H_t P_{d-} + P_{d-}H_t P_{s0}). \tag{A.9}
$$

It satisfies $[S, UC + \mu M_a] = -V$, so we turn to second-order corrections computed from the power-series expansion of the Schrieffer-Wolff transformation: $\delta H = \frac{1}{2}P_{s0}[S, V]P_{s0}$. In particular, we find

$$
\delta H_{\text{BH,eff}} = -\frac{1}{\mu}(-P_{s0}H_t P_{s+}H_t P_{s0} + P_{s0}H_t P_{s-}H_t P_{s0})
$$

$$
- \frac{1}{U+\mu}P_{s0}H_t P_{d+}H_t P_{s0} - \frac{1}{U-\mu}P_{s0}H_t P_{d-}H_t P_{s0}. \tag{A.10}
$$

Let us now go term a term. We note the first term describes a hard-core process, as the one considered in the Rydberg case. Apart from a normalizing factor of two, the key distinction comes from the phase dressing in Eq. (14). When we expand the formula for the hopping, we obtain

$$
\delta H_{\text{BH,eff}}^{(\mu)} = \frac{t^2}{\mu}\sum_{ij}{}'(-1)^{i+j}\Big[(\sigma_{2i-1}^+ + \sigma_{2i+1}^+)\sigma_{2i}^-, \sigma_{2j}^+(\sigma_{2j-1}^- + \sigma_{2j+1}^-)\Big], \tag{A.11}
$$

where we use the $P_{s0}$ projection to replace boson for spin-1/2 operators. From the commutator, we then find

$$
\delta H_{\text{BH,eff}}^{(\mu)} = -\frac{t^2}{\mu}\sum_i(\sigma_i^+ \sigma_{i+1}^z \sigma_{i+2}^- + \text{H.c.}), \tag{A.12}
$$

where we drop additive constants. The other two terms in the second line of Eq. (A.10), come from virtual processes that contain a doubly occupied site. When the doubly-occupied site happen to be in the even sublattice, we get

$$
\delta H_{\text{BH,eff}}^{(U+\mu)} = -\frac{t^2}{U+\mu}\sum_i{}'\Big[\sigma_{2i-1}^+(1+\sigma_{2i}^z)\sigma_{2i+1}^- + \sigma_{2i-1}^-(1+\sigma_{2i}^z)\sigma_{2i+1}^+\Big]
$$

$$
- \frac{t^2}{2(U+\mu)}\sum_i{}'(\sigma_{2i-1}^z + \sigma_{2i+1}^z)\sigma_{2i}^z. \tag{A.13}
$$

Likewise, when the virtual doubly-occupied site belongs to the odd sublattice,

$$
\delta H_{\text{BH,eff}}^{(U-\mu)} = +\frac{t^2}{U-\mu}\sum_i{}'\Big[\sigma_{2i}^+(1+\sigma_{2i+1}^z)\sigma_{2i+2}^- + \sigma_{2i}^-(1+\sigma_{2i+1}^z)\sigma_{2i+2}^+\Big]
$$

$$
- \frac{t^2}{2(U-\mu)}\sum_i{}'(\sigma_{2i}^z + \sigma_{2i+2}^z)\sigma_{2i+1}^z. \tag{A.14}
$$

Collecting results from Eqs. (A.12), (A.13) and (A.14), we arrive at the effective Hamiltonian (15). Note that the doubly-occupied processes have different couplings for elements in the odd and even sublattices.

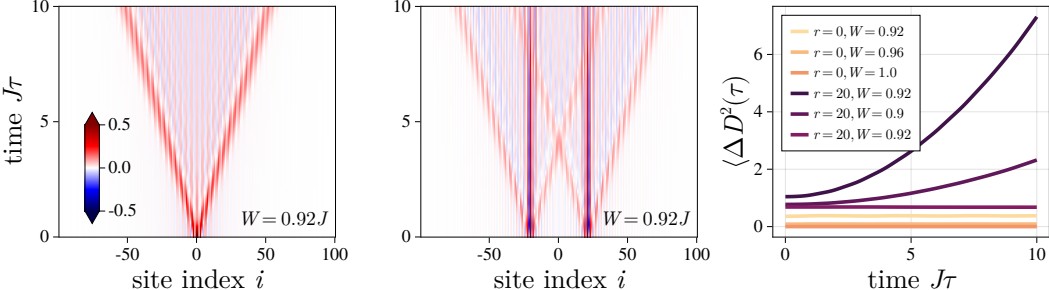

Figure 12: Two-particle dynamics in the dipole TLL state. (a) Time evolution of the density variation when two defects are created in consecutive sites. (b) Time evolution of the density variation when defects are created far apart. (c) Variance of dipole as a function of time for a local pair ($r = 0$) and a separated pair ($r = 20$).

## B Dynamics of an excitation pair

In this appendix we report results obtained for the quench dynamics of a pair of defects. The two excitations are created upon the action of $\sigma_i^+ \sigma_j^+$ onto the half-filled ground state. The numerical details follow the prescription of Sec. 5.

We remain close to the strictly confined point, i.e., $|W-J|$ small, and consider two different scenarios. In the first case we create a pair in adjacent sites $\sigma_i^+ \sigma_{i+1}^+$, while in the second case we take them far apart $\sigma_{i-r}^+ \sigma_{i+1+r}^+$, with $r = 20$. Figure 12 shows the results obtained for a chain with $L = 200$ sites. In Fig. 12(a) we observe a clear light-cone signal formed upon the time evolution of a creation of the local pair. Note that the ballistic spreading of the local pair is in agreement with the continuum limit field theory, featuring a motion compatible with the emergent dipole symmetry. This is further showcased in Fig. 12(c), where we observe that the variance of the dipole does not evolve significantly in time; cf. the variance for a single defect in Fig. 8(d). Finally, in Fig. 12(b) a separated pair exhibits the approximate behavior of independent single-particle defects. This is further reinforced in Fig. 12(c) where we see the dipole variance changes with time whenever $|W-J| \neq 0$.

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
