# Peer review of "Emergent dipole field theory in atomic ladders"

_SciPost Physics_

## Round 1 · Referee Report · Anonymous (Referee 1) · 2024-10-23

Report

The manuscript is well written and organized and meets the criteria for publication in SciPost. Before publication there are required revisions .
i) The authors use the Tomonaga-Luttinger liquid (TLL) model showing a dipole symmetry and discuss quench dynamics. They should stress the conditions under which the low-energy effective field theory is still applicable under a quench dynamics. They could also refer to papers studying the quench dynamics of one dimensional Bose gases.
ii) The numerical simulations are done by considering a finite size system so the authors should discuss under which condition the field theory description, i.e. the continuum picture, is compatible with the numerical simulations.
iii) The last sentence of the abstract " We present a blueprint protocol to observe the effect of emergent dipole symmetry in such experimental platforms, combining adiabatic state preparation with quench dynamics." is incomprehensible, specially what is meant by combination of adiabatic state preparation with quench dynamics.

Recommendation

Ask for major revision

---

## Round 1 · Referee Report · Anonymous (Referee 2) · 2024-11-7

Report

The authors have studied the physics of hardcore bosons in zig-zag ladder with correlated hopping and Ising type interaction. They have obtained the phase diagram using the analytical (bosonization) and numerical (ED) approaches. The main finding is the appearance of a paired Tomonaga-Luttinger liquid (TLL) phase which features an emergent dipole symmetry. This effect has been examined using the quench dynamics of particle defects. They have also proposed experimental realization of the model and present a protocol to observe the effect of the emergent dipole symmetry.

In general the paper contains a detailed analysis of the phases and the transitions and the underlying physics. The calculations (both numerical and analytical) seem concrete. While I feel that the model and the main result i.e. the appearance of the pair TLL phase is not very new, the approach that explains the appearance of such phase is certainly new as compared to earlier studies. They have demonstrated and explained signatures of various phases and phase transitions in the parameter space using various order parameters computed using numerics. In my opinion the manuscript is suitable for publication in Scipost physics. However, before the acceptance I would like the authors to clarify some of the important points which I feel would enhance some of the missing clarity in the manuscript.

-- The statement in the last paragraph in page-4 which says that the PS phases are sensitive to the form of interactions is not clear to me. Some clarification on this will be helpful.

-- The statement that the bosons in one leg will move when assisted by a partner in the neighboring chain (I think it should be leg instead of chain) is not clear from Fig.4. It seems to me from Fig. 4 that the dimer actually breaks rather than moving together.

-- Some clarification related to the selection of the states for different signs of W in Sec. 2.1 should be added.

-- It is not well explained on why the appearance of the diagonal terms in Fig. 7 ensures the emergent dipole symmetry. Can data for W=J be given in Fig. 7? Or is there any specific reason behind excluding this result.

-- Reference to Kramers-Wannier transformation should be given.

-- In the results shown Fig.7, only V=0 case has been considered. Is there any specific reason for this?

-- Do the variation of different parameters in Fig. 8 (a) follow a certain functional form or are they just linearly varied?

-- What is the significance of showing Fig. 8 (b) and highlighting that the low-energy manifold remains separated from the rest of the spectrum? What does the color code in Fig. 8(b) indicate?

-- I am not very sure how reliable the single and two particle correlations are as far as ED calculations are considered.

Recommendation

Publish (meets expectations and criteria for this Journal)

---

## Round 2 · Referee Report · Anonymous (Referee 2) · 2025-2-13

Report

The authors have addressed to the queries and comments raised by me. The response and the modifications to the manuscript are satisfactory. I can now recommend the manuscript for publication.

Recommendation

Publish (easily meets expectations and criteria for this Journal; among top 50%)

---

## Round 2 · Author Response

Dear Editor,

We hereby resubmit a revised version of our manuscript. We thank the Referees for their feedback, and implemented changes in the text accordingly. We append below a point-by-point reply to each one of the questions raised in the reports.

Best regards,

HB Xavier, PS Tarabunga, M Dalmonte, and RG Pereira

---

## Round 2 · List of Changes

* * *
Response to Report 1
* * *
i) The authors use the Tomonaga-Luttinger liquid (TLL) model showing a dipole symmetry and discuss quench dynamics. They should stress the conditions under which the low-energy effective field theory is still applicable under a quench dynamics. They could also refer to papers studying the quench dynamics of one dimensional Bose gases.

1. We do not expect TLL theory to be arbitrarily applicable to quenches. The field theory should be applicable to the quench dynamics as long as the latter is dictated by low-energy excitations. These are the types of quenches we consider in our work. As the action of \sigma_j^\pm also creates a defect above the gap, we include its effect via the heavy mobile impurity picture. We have added a sentence in sec 5.1 to clarify this point. References [55] and [56], which study the dynamics of one-dimensional Bose gases, have been added to the bibliography.

ii) The numerical simulations are done by considering a finite size system so the authors should discuss under which condition the field theory description, i.e. the continuum picture, is compatible with the numerical simulations.

2. The continuum field theory description is only valid in the bulk of sufficiently large systems. Given the characteristic velocity v_c of charge excitations, the distance d to the edges sets a time scale T ~ d/v_c for which finite-size effects become appreciable. We minimize these effects by stopping simulations whenever the fastest light-cone comes close to the edges, see for instance Figs. 9, 11, and 12. We have added a comment on this in Sec. 5.3.

iii) The last sentence of the abstract " We present a blueprint protocol to observe the effect of emergent dipole symmetry in such experimental platforms, combining adiabatic state preparation with quench dynamics." is incomprehensible, specially what is meant by combination of adiabatic state preparation with quench dynamics.

3. Last sentence of the abstract has been revised.
* * *
Response to Report 2
* * *
-- The statement in the last paragraph in page-4 which says that the PS phases are sensitive to the form of interactions is not clear to me. Some clarification on this will be helpful.

1. This point can be understood as follows. First, the PS phases feature spatial modulations comparable to system size, being strongly dependent on boundary fields, as well as the change from open to closed boundary conditions. Since the mapping from spin to density operators involves \sigma_i^z= 2n_i -1, the change from Ising-like interactions to density-density affects the chemical potential, which for open boundary conditions leaves behind extra boundary fields. We clarify this point in the updated manuscript.

-- The statement that the bosons in one leg will move when assisted by a partner in the neighboring chain (I think it should be leg instead of chain) is not clear from Fig.4. It seems to me from Fig. 4 that the dimer actually breaks rather than moving together.

2. We have revised Fig. 4 and its caption for clarity. We stresss that the dimer is formed by two bosons on neighboring sites in different legs and can change its orientation as the particles move, but the two particles remain together.

-- Some clarification related to the selection of the states for different signs of W in Sec. 2.1 should be added.

3. The final paragraph of Sec 2.1 has been revised to increase clarity. The key idea is that states \psi_1 and \psi_2 differ from a particle-hole transformation, broken by the W coupling. For W>0 the particle dimer in the center of \psi_1 is able to lower its energy, and vice-versa for W<0.

-- It is not well explained on why the appearance of the diagonal terms in Fig. 7 ensures the emergent dipole symmetry. Can data for W=J be given in Fig. 7? Or is there any specific reason behind excluding this result.

4. We added an explanation to the last paragraph of Sec. 5.1. The case W=J is left out because it is purely diagonal, being very similar to the case W=0.9J already displayed in Fig. 7.

-- Reference to Kramers-Wannier transformation should be given.

5. We have added citation to Ref. [40].

-- In the results shown Fig.7, only V=0 case has been considered. Is there any specific reason for this?

6. The addition of dipole-preserving perturbations, such as the V-interaction does not change results in Fig. 7. This point is now stressed in the last paragraph of Sec. 5.1.

-- Do the variation of different parameters in Fig. 8 (a) follow a certain functional form or are they just linearly varied?

7. Indeed, as the Referee points out, they are just linearly varied. This is a simple way to evolve the initial parameters to the desired ones, and we show that the resulting state preparation already performs well. This of course still leaves open the possibility to search for better optimized parameter ramps.

-- What is the significance of showing Fig. 8 (b) and highlighting that the low-energy manifold remains separated from the rest of the spectrum? What does the color code in Fig. 8(b) indicate?

8. Figure 8(b) shows the spectrum in the static limit, which is equivalent to the case of an infinitely long, truly adiabatic preparation. If level crossings occur within this limit, the dynamical preparation of the target state becomes more challenging. In particular, the fact that the ground state does not exhibit crossing means that the state at the final time is the ground state at the given parameter. The color code was just a guide to the eye for different energy levels. In the updated version of the manuscript the color code was removed to avoid further misunderstandings.

-- I am not very sure how reliable the single and two particle correlations are as far as ED calculations are considered.

9. Although not representative of true thermodynamic limit for the sizes considered, we compare dynamical TDVP results to ED calculations as they are accurate predictions for the target model in the static limit. Also, for state preparation, we chose to focus on relatively small volumes as those are likely relevant to first experiments, so the correlations indicate whether capturing the qualitative physics is possible within a given time scale.

---

## Editorial Decision

in_voting